# Submodular Maximization Through Barrier Functions

**Ashwinkumar Badanidiyuru**
Google
ashwinkumarbv@gmail.com

**Amin Karbasi**
Yale University
amin.karbasi@yale.edu

**Ehsan Kazemi**
Google
ehsankazemi@google.com

**Jan Vondrák**
Stanford University
jvondrak@stanford.edu

## Abstract

In this paper, we introduce a novel technique for constrained submodular maximization, inspired by barrier functions in continuous optimization. This connection not only improves the running time for constrained submodular maximization but also provides the state of the art guarantee. More precisely, for maximizing a monotone submodular function subject to the combination of a $k$-matchoid and $\ell$-knapsack constraints (for $\ell \leq k$), we propose a potential function that can be approximately minimized. Once we minimize the potential function up to an $\varepsilon$ error, it is guaranteed that we have found a feasible set with a $2(k+1+\varepsilon)$-approximation factor which can indeed be further improved to $(k+1+\varepsilon)$ by an enumeration technique. We extensively evaluate the performance of our proposed algorithm over several real-world applications, including a movie recommendation system, summarization tasks for YouTube videos, Twitter feeds and Yelp business locations, and a set cover problem.

## 1 Introduction

In the constrained continuous optimization, barrier functions are usually used to impose an increasingly large cost on a feasible point as it approaches the boundary of the feasible region [40]. In effect, barrier functions replace constraints by a penalizing term in the primal objective function so that the solution stays away from the boundary of the feasible region. This is an attempt to approximate a constrained optimization problem with an unconstrained one and to later apply standard optimization techniques. While the benefits of barrier functions are studied extensively in the continuous domain [40], their use in discrete optimization is not very well understood.

In this paper, we show how discrete barrier functions manifest themselves in constrained submodular maximization. Submodular functions formalize the intuitive diminishing returns condition, a property that not only allows optimization tractability but also appears in many machine learning applications, including video, image, and text summarization [16, 31, 45], active set selection in non-parametric learning [34], sensor placement and information gathering [14]. Formally, for a ground set $\mathcal{N}$, a non-negative set function $f : 2^{\mathcal{N}} \rightarrow \mathbb{R}_{\geq 0}$ is **submodular** if for all sets $A \subseteq B \subset \mathcal{N}$ and every element $e \in \mathcal{N} \setminus B$, we have $f(A \cup \{e\}) - f(A) \geq f(B \cup \{e\}) - f(B)$. The submodular function $f$ is monotone if for all $A \subseteq B$ we have $f(A) \leq f(B)$.

The celebrated results of Nemhauser et al. [39] and Fisher et al. [12] show that the vanilla greedy algorithm provides an optimal approximation guarantee for maximizing a monotone submodular function subject to a cardinality constraint. However, the performance of the greedy algorithm degrades as the feasibility constraint becomes more complex. For instance, the greedy algorithm does

not provide any constant factor approximation guarantee if we replace the cardinality constraint with a knapsack constraint. Even though there exist many works that achieve the tight approximation guarantee for maximizing a monotone submodular function subject to multiple knapsack constraints, the running time of these algorithms is prohibitive as they either rely on enumerating large sets or running the continuous greedy algorithm. In contrast, we showcase a fundamentally new optimization technique through a **discrete barrier function minimization** in order to efficiently handle knapsack constraints and develop fast algorithms. More formally, we consider the following constrained submodular maximization problem defined over the ground set $\mathcal{N}$:

$$S^* = \operatorname*{argmax}_{\substack{S \subseteq \mathcal{N},\, S \in \mathcal{I} \\ c_i(S) \leq 1 \,\forall\, i \in [\ell]}} f(S) \;, \tag{1}$$

where the constraint is the intersection of a $k$-matchoid constraint $\mathcal{M}(\mathcal{N}, \mathcal{I})$ (a general subclass of $k$-set systems) and $\ell$ knapsacks constraints $c_i$ (for $i \in [\ell]$). We assume that both the objective function $f$ and the constraints are accessed through a *value oracle* and a *membership oracle*, respectively [5]. We measure the complexity of an algorithm by the number of value oracle queries it performs, as in most cases, value oracle queries dominate the actual time complexity of the algorithm [5, 26].

**Contributions.** We propose two algorithms for maximizing a monotone submodular function subject to the intersection of a $k$-matchoid and $\ell$ knapsack constraints. Our approach uses a novel barrier function technique and lies in between fast thresholding algorithms with suboptimal approximation ratios and slower algorithms that use continuous greedy and rounding methods. The first algorithm, BARRIER-GREEDY, obtains a $2(k + 1 + \varepsilon)$-approximation ratio with $\tilde{O}(nr + r^3)$ value oracle calls, where $r$ is the maximum cardinality of a feasible solution.[1] The second algorithm, BARRIER-GREEDY++, obtains a better approximation ratio of $(k + 1 + \varepsilon)$, but at the cost of $\tilde{O}(n^3 r + n^2 r^3)$ value oracle calls. BARRIER-GREEDY is theoretically fast and even exhibits better performance in practice while achieving a near-optimal approximation ratio. The main purpose of BARRIER-GREEDY++ is to provide a tighter approximation guarantee at the expense of a higher computational cost. Indeed, there is a trade-off between the quality of the solution we desire to obtain (higher for BARRIER-GREEDY++) and the time we are willing to spend (faster for BARRIER-GREEDY). In our experiments, we opted for BARRIER-GREEDY which is the more scalable method. Our results show that barrier function minimization techniques provide a versatile algorithmic tool for constrained submodular optimization with strong theoretical guarantees that may scale to many previously intractable problem instances. We demonstrate the practical effectiveness of our algorithms over **several real-world machine learning applications**, including a movie recommendation system, summarization tasks for YouTube videos, Twitter feeds of news agencies and Yelp business locations, and a set cover problem. **Proofs are deferred to the Supplementary Material.**

## 2 Related Work

The problem of maximizing a monotone submodular function subject to various constraints goes back to the seminal work of Nemhauser et al. [39] and Fisher et al. [12] which showed that the greedy algorithm gives a $(1 - 1/e)^{-1}$-approximation subject to a cardinality constraint, and more generally a $(k+1)$-approximation for any $k$-system (which subsumes the intersection of $k$ matroids, and also the $k$-matchoid constraint considered here). Nemhauser and Wolsey [38] showed that the factor of $(1 - 1/e)^{-1}$ is best possible in this setting. After three decades, there was a resurgence of interest in this area due to new applications in economics, game theory, and machine learning. While we cannot do justice to all the work that has been done in submodular maximization, let us mention the works most relevant to ours—in particular focusing on matroid/matchoid and knapsack constraints.

Sviridenko [43] gave the first algorithm to achieve a $(1 - 1/e)^{-1}$-approximation for submodular maximization subject to a knapsack constraint. This algorithm, while relatively simple, requires enumeration over all triples of elements and hence its running time is rather slow ($\tilde{O}(n^5)$). Vondrák [46] and Călinescu et al. [6] gave the first $(1 - 1/e)^{-1}$-approximation for submodular maximization subject to a matroid constraint. This algorithm, continuous greedy with pipage rounding, is also

relatively slow (at least $\tilde{O}(n^3)$ value oracle calls). Using related techniques, Kulik et al. [27] gave a $(1 - 1/e)^{-1}$-approximation subject to any constant number of knapsack constraints, and Chekuri et al. [7] gave a $(1 - 1/e)^{-1}$-approximation subject to one matroid and any constant number of knapsack constraint; however, these algorithms are even slower and less practical.

Following these results (optimal in terms of approximation), applications in machine learning called for more attention being given to running time and practicality of the algorithms (as well as other aspects, such as online/streaming inputs and distributed/parallel implementations, which we do not focus on here). In terms of improved running times, Gupta et al. [15] developed fast algorithms for submodular maximization (motivated by the online setting), however with suboptimal approximation factors. Badanidiyuru and Vondrák [1] provided a $(1 - 1/e - \varepsilon)^{-1}$-approximation subject to a cardinality constraint using $O(\frac{n}{\varepsilon} \log \frac{1}{\varepsilon})$ value oracle calls, and subject to a matroid constraint using $O(\frac{n^2}{\varepsilon^4} \log^2 \frac{n}{\varepsilon})$ value oracle calls. Also, they gave a fast thresholding algorithm providing a $(k + 2\ell + 1 + \varepsilon)$-approximation for a $k$-system combined with $\ell$ knapsack constraints using $O(\frac{n}{\varepsilon^2} \log^2 \frac{n}{\varepsilon})$ value oracle calls. This was further generalized to the non-monotone setting [33].

The approximation factor of $k + 1$ for BARRIER-GREEDY++ matches the greedy algorithm for $k$ matroid constraints (without the knapsack constraints) [12]. The only known improvement of this result requires a more sophisticated (and very slow) local-search algorithm [29]. In terms of provable hardness, it is known that it is NP-hard to achieve a better than $k/(\log k)$-approximation for $k$-dimensional matching [19], which is a (very) special case of our problem. Generally, it is believed that the hardness for this class of problems to be in the order of $\Theta(k)$.

Submodular maximization has numerous real-world applications, including sensor placement and information gathering [4, 14], recommendation systems [9, 37, 41] and data summarization [16, 22, 24, 25, 31, 36, 44, 45]. Recent works have studied the applications of constrained submodular maximization in scenarios similar to setting of this paper. Mirzasoleiman et al. [35] and Feldman et al. [11] cast video and location data summarization applications to a submodular maximization problem subject to $k$-matchoid constraints. Feldman et al. [10] and Haba et al. [17] studied a movie recommendation system with a $k$-extendible constraint. Mirzasoleiman et al. [33] used the intersection of $k$ matroids and $\ell$ knapsacks to model recommendation systems, image summarization and revenue maximization tasks.

## 3 Preliminaries and Notation

Let $f : 2^{\mathcal{N}} \to \mathbb{R}_+$ be a non-negative and monotone submodular function defined over ground set $\mathcal{N}$. Given an element $a$ and a set $A$, we use $A + a$ as a shorthand for the union $A \cup \{a\}$. We also denote the marginal gain of adding $a$ to a $A$ by $f(a \mid A) \triangleq f(A + a) - f(A)$. Similarly, the marginal gain of adding a set $B \subseteq \mathcal{N}$ to another set $A \subseteq \mathcal{N}$ is denoted by $f(B \mid A) \triangleq f(B \cup A) - f(A)$.

A *set system* $\mathcal{M} = (\mathcal{N}, \mathcal{I})$ with $\mathcal{I} \subseteq 2^{\mathcal{N}}$ is an *independence system* if $\varnothing \in \mathcal{I}$ and $A \subseteq B$, $B \in \mathcal{I}$ implies that $A \in \mathcal{I}$. In this regard, a set $A \in \mathcal{I}$ is called independent, and a set $B \notin \mathcal{I}$ is called dependent. A *matroid* is an independence system with the following additional property: if $A$ and $B$ are independent sets obeying $|A| < |B|$, then there exists $a \in B \setminus A$ such that $A + a$ is independent. In this paper, we consider two different constraints. The first constraint is in an intersection of $k$ matroids or a $k$-matchoid (as a generalization of the intersection of $k$-matroids). The second constraint is the set of $\ell$ knapsacks for $\ell \leq k$. Next, we formally define these constraints.

**Definition 3.1.** *Let* $\mathcal{M}_1 = (\mathcal{N}, \mathcal{I}_1), \dots, \mathcal{M}_k = (\mathcal{N}, \mathcal{I}_k)$ *be* $k$ *matroids over the ground set* $\mathcal{N}$. *An intersection of* $k$ *matroids is an independent system* $\mathcal{M}(\mathcal{N}, \mathcal{I})$ *such that* $\mathcal{I} = \{S \subseteq \mathcal{N} \mid \forall i, S \in \mathcal{I}_i\}$.

**Definition 3.2.** *An independence set system* $(\mathcal{N}, \mathcal{I})$ *is a* $k$-*matchoid if there exist* $m$ *different matroids* $(\mathcal{N}_1, \mathcal{I}_1), \dots, (\mathcal{N}_m, \mathcal{I}_m)$ *such that* $\mathcal{N} = \cup_{i=1}^m \mathcal{N}_i$, *each element* $e \in \mathcal{N}$ *appears in no more than* $k$ *ground sets among* $\mathcal{N}_1, \dots, \mathcal{N}_m$ *and* $\mathcal{I} = \{S \subseteq \mathcal{N} \mid \forall i, \mathcal{N}_i \cap S \in \mathcal{I}_i\}$.

A knapsack constraint is defined by a cost vector $c$ for the ground set $\mathcal{N}$, where for the cost of a set $S \subseteq \mathcal{N}$ we have $c(S) = \sum_{e \in S} c_e$. Given a knapsack capacity (or budget) $C$, a set $S \subseteq \mathcal{N}$ is said to satisfy the knapsack constraint $c$ if $c(S) \leq C$. We assume, without loss of generality, the capacity of all knapsacks $c_i$ (for $1 \leq i \leq \ell$) are normalized to 1. We denote the cost of element $e$ under the $i$th knapsack by $c_{i,e}$.

Assume there is a global ordering of elements $\mathcal{N} = \{1, 2, 3, \ldots, |\mathcal{N}|\}$. For a set $S \subseteq \mathcal{N}$ and an element $a \in \mathcal{N}$, the benefit (or contribution) of $a$ to $S$ (denoted by $w_{S,a}$) is defined as follows: (i) If $a \in S$: $w_{S,a}$ is the marginal gain of adding $a$ to all elements of $S$ that are smaller than $a$, i.e., $w_{S,a} = f(S \cap [a]) - f(S \cap [a-1])$. (ii) If $a \notin S$: $w_{S,a}$ is the marginal gain of adding $a$ to $S$, i.e., $w_{S,a} = f(S + a) - f(S)$. From the submodularity of $f$, it is straightforward to show that $f(S) = \sum_{a \in S} w_{S,a}$. Furthermore, for each element $a$, $\gamma_a = \sum_{i=1}^{\ell} c_{i,a}$ represents the aggregate cost of $a$ over all knapsacks. It is easy to see that $\sum_{i=1}^{\ell} c_i(S) = \sum_{a \in S} \gamma_a$. We denote the latter quantity, the aggregate cost of all elements of $S$ over all knapsacks, by $\gamma(S)$. Since the capacity of each one of the $\ell$ knapsacks is normalized to 1, for any feasible solution $S$, we have always $\gamma(S) \leq \ell$.

# 4  The Barrier Function and Our Algorithms

In this section, we first explain our proposed barrier function. We then present BARRIER-GREEDY and BARRIER-GREEDY++ and prove that these two algorithms, by finding a local minimum of the barrier function, can efficiently maximize a monotone submodular function subject to the intersection of $k$-matroids and $\ell$ knapsacks. As a generalization of our results, In Appendix C, we show how our algorithms could be extended to $k$-matchoids and $\ell$ knapsack constraints.

## 4.1  The BARRIER-GREEDY Algorithm

Existing local search algorithms under $k$ matroid constraints try to maximize the objective function over a space of $O(n^k)$ feasible swaps [28, 29]. Generally, the addition of knapsack constraints makes the structure of feasible swaps even more complicated. Our proposed method, a new **local-search** algorithm called BARRIER-GREEDY, avoids the exponential dependence on $k$ while it incorporates the additional knapsack constraints.

As a first technical contribution, instead of making the space of feasible swaps huge and more complicated, we incorporate the knapsack constraints into a **potential function** similar to **barrier functions** in the continuous optimization domain. For a set function $f(S)$ and the intersection of $k$ matroids and $\ell$ knapsack constraints $c_i(S) \leq 1, i \in [\ell]$, we propose the following potential function:

$$\phi(S) = \frac{\mathtt{OPT} - (k+1)f(S)}{1 - \sum_{i=1}^{\ell} c_i(S)} \quad, \tag{2}$$

where $\mathtt{OPT}$ is the optimum value for Problem (1). This potential function involves the knowledge of $\mathtt{OPT}$—we replace this by an estimate that we can "guess" (enumerate over) efficiently by a standard technique explained later. Intuitively, Eq. (2) turns the difficult knapsack constraint into a smooth objective function. The particular form of our potential function ensures that getting close to infeasible sets is penalized, while the function is still simple to deal with as an objective function. This potential function treats the knapsack constraints in a very conservative way: while $\gamma(S)$ for a feasible set $S$ could be as large as $\ell$, we consider only sets with $\gamma(S) \leq 1$, whereas for sets with a larger weight the potential function becomes negative.[2]

A technique first proposed by Badanidiyuru et al. [2] can be used to guess the optimum value $\mathtt{OPT}$. We denote this guess by $\Omega$. From the submodularity of $f$, we can deduce that $M \leq \mathtt{OPT} \leq rM$, where $M$ is the largest value in the set $\{f(\{j\}) \mid j \in \mathcal{N}\}$ and $r$ is the maximum cardinality of a feasible solution. Then, it suffices to try $O(\frac{\log r}{\varepsilon})$ different guesses in the set $\Lambda = \{(1 + \varepsilon)^i \mid {}^M/_{(1+\varepsilon)} \leq (1 + \varepsilon)^i \leq rM\}$ to obtain a close enough estimate of $\mathtt{OPT}$. In the rest of this section, we assume that we have access to a value of $\Omega$ such that $(1 - \varepsilon)\mathtt{OPT} \leq \Omega \leq \mathtt{OPT}$. Using $\Omega$ as an estimate of $\mathtt{OPT}$, our potential function converts to

$$\phi(S) = \frac{\Omega - (k+1)f(S)}{1 - \gamma(S)} \quad. \tag{3}$$

The main goal of BARRIER-GREEDY is to efficiently **minimize** the potential function (3) in several consecutive sequential rounds. This potential function is designed in a way such that either the

current solution respects all the knapsack constraints or if the solution violates any of the knapsack constraints, we can guarantee that the objective value is already sufficiently large.

As a second technical contribution, we optimize the local search procedure for $k$ matroids. More precisely, we improve the previously known $O(n^k)$ running time of Lee et al. [28] to a new method with $\tilde{O}(nr + r^3)$ value oracle calls. This is accomplished by a novel greedy approach that efficiently searches for the best existing swap, instead of a brute-force search among all possible swaps. With these two points in mind, we proceed to explain our first proposed algorithm BARRIER-GREEDY.

To quantify the effect of each element $a$ on the potential function $\phi(S)$, as a notion of their individual energy, we define the following quantity:

$$\delta_{S,a} = (k+1)(1 - \gamma(S))w_{S,a} - (\Omega - (k+1)f(S))\gamma_a \ . \tag{4}$$

The quantity $\delta_{S,a}$ measures how desirable an element $a$ is for set $S$, i.e., larger values of $\delta_{S,a}$ would have a larger effect on the potential function. Also, the potential function is designed such that any $a \in S$ with $\delta_{S,a} \leq 0$ can be removed from $S$ without increasing the potential function.

BARRIER-GREEDY starts with an empty set $S$ and performs the following two steps for at most $r \log(1/\varepsilon)$ iterations or till it reaches a solution $S$ such that $f(S) \geq \frac{(1-\varepsilon)\Omega}{k+1}$: Firstly, it finds an element $b \in \mathcal{N} \setminus S$ with the maximum value of $\delta_{S,b} - \sum_{i \in J_b} \delta_{S,a_i}$ such that $S - a_i + b \in \mathcal{I}_i$ for $a_i \in S$ and $i \in J_b \triangleq \{j \in [k] : S + b \notin \mathcal{I}_j\}$. If $J_b$ is an empty set, adding $b$ would not violate any of the matroid constraints and we need to only consider the contribution of $\delta_{S,b}$. In this step, we need to compute $\delta_{S,a}$ for all elements $a \in \mathcal{N}$ only once and then we can use these pre-computed values to find the best candidate $b$. The outcome of this step is to find an element $b$ such that its addition to set $S$ and removal of a corresponding set of elements from $S$ decrease the potential function by a large margin while still keeping the solution feasible. In the second step, BARRIER-GREEDY removes all elements with $\delta_{S,a} \leq 0$ form set $S$. In Lemma A.1, we prove that these removals could only decrease the potential function.

BARRIER-GREEDY finds a solution with a good objective value mainly for two reasons: (i) If it continues for $r \log(1/\varepsilon)$ iterations, we prove that the potential function would be very close to 0, which consequently enables us to guarantee the performance for this case. Note that, for our solution, we maintain the invariant that $\gamma(S) < 1$ to make sure the knapsack constraints are also satisfied. (ii) If $f(S) \geq \frac{(1-\varepsilon)\Omega}{k+1}$, we prove that the objective value of one of the two feasible sets $S \setminus \{b\}$ and $\{b\}$ is at least $\frac{(1-\varepsilon)\Omega}{2(k+1)}$, where $b$ is the last added element to $S$. The details of BARRIER-GREEDY are described in Algorithm 1. Theorem 4.1 guarantees the performance of BARRIER-GREEDY. The proof for Theorem 4.1 is given in Appendix A.

---

**Algorithm 1** BARRIER-GREEDY
___
**Input:** $f : 2^{\mathcal{N}} \to \mathbb{R}_+$, membership oracles for $k$ matroids $\mathcal{M}_1 = (\mathcal{N}, \mathcal{I}_1), \ldots, \mathcal{M}_k = (\mathcal{N}, \mathcal{I}_k)$, and $\ell$ knapsack-cost functions $c_i : \mathcal{N} \to [0, 1]$.
**Output:** A set $S \subseteq \mathcal{N}$ satisfying $S \in \bigcap_{i=1}^{k} \mathcal{I}_i$ and $c_i(S) \leq 1 \ \forall i$.
   $M \leftarrow \max_{j \in \mathcal{N}} f(\{j\})$
   $\Lambda \leftarrow \{(1+\varepsilon)^i \mid M/(1+\varepsilon) \leq (1+\varepsilon)^i \leq rM\}$       ▷ *Candidate guesses for estimating* OPT.
   **for** $\Omega \in \Lambda$ **do**
      $S \leftarrow \varnothing$                        ▷ *For each guess* $\Omega$, *we find a soltion* $S$.
      **while** $f(S) < \frac{1-\varepsilon}{k+1}\Omega$ and iteration number is smaller than $r \log \frac{1}{\varepsilon}$ **do**
         Find $b \in \mathcal{N} \setminus S$ and $a_i \in S$ for $i \in J_b = \{j \in [k] : S + b \notin \mathcal{I}_j\}$ such that $S - a_i + b \in \mathcal{I}_i$
         and $\delta_{S,b} - \sum_{i \in J_b} \delta_{S,a_i}$ is maximized       ▷ *Compute* $\delta_{S,a}$ *from Eq.* (4).
         $S \leftarrow S \setminus \{a_i : i \in J_b\} + b$
         **while** $\exists a \in S$ s.t. $\delta_{S,a} \leq 0$ **do** Remove $a$ from $S$
      **if** set $S$ satisfies all the knapsack constraints **then**
         $S_\Omega \leftarrow S$
      **else**
         $S_\Omega \leftarrow \operatorname{argmax}\{f(\{b\}), f(S - b)\}$, where $b$ is the last added element to $S$
   **return** $\operatorname{argmax}_{\Omega \in \Lambda} f(S_\Omega)$
___

**Theorem 4.1.** BARRIER-GREEDY *(Algorithm 1) provides a* $2(k+1+\varepsilon)$*-approximation for maximizing a monotone submodular function subject to the intersection of* $k$ *matroids and* $\ell$ *knapsack constraints (for* $\ell \leq k$*). It performs* $O(\frac{nr+r^3}{\varepsilon} \log r \log 1/\varepsilon)$ *value oracle calls and* $O(\frac{nkr^2}{\varepsilon} \log r \log 1/\varepsilon)$ *membership oracle calls, where* $r$ *is the maximum cardinality of a feasible solution.*

Our proofs show that, in Theorem 4.1 (and later in Theorem 4.2), we can drop the assumption $\ell \leq k$ and replace $k$ in the approximation factors with the bigger value between $k$ and $\ell$.

BARRIER-GREEDY, as we mentioned above, considers only sets $S$ where the sum of all the $\ell$ knapsacks is at most 1 for them, i.e., sets $S$ such that $\gamma(S) = \sum_i^\ell \sum_{a \in S} c_{i,a} \leq 1$. For scenarios with more than one knapsack, while BARRIER-GREEDY theoretically produces a highly competitive objective value, there might be feasible solutions such that they fill the capacity of all knapsacks, i.e., $\gamma(S)$ could be very close to $\ell$ for them. While our proposed algorithms fail to find these kinds of solutions, in Appendix D, inspired by our theoretical results, we design a heuristic algorithm (called BARRIER-HEURISTIC) that overcomes this issue.

## 4.2 The BARRIER-GREEDY++ Algorithm

In this section, we use an enumeration technique to improve the approximation factor of BARRIER-GREEDY to $(k + 1 + \varepsilon)$. For this reason, we propose the following modified algorithm: for each feasible pair of elements $\{a', a''\}$, define a reduced instance where the objective function $f$ is replaced by a monotone submodular function $g(S) \triangleq f(S \cup \{a', a''\}) - f(\{a', a''\})$, and the knapsack capacities are decreased by $c_{i,a'} + c_{i,a''}$. In this reduced instance, we remove the two elements $a', a''$ and all elements $a \in \mathcal{N} \setminus \{a', a''\}$ with $g(\{a\}) > \frac{1}{2} f(\{a', a''\})$ from the ground set $\mathcal{N}$. In the reduced instance, we consider contractions of all the $k$ matroids to set $\{a', a''\}$ as the new set of matroid constraints. All the elements $a$ with $g(\{a\}) > \frac{1}{2} f(\{a', a''\})$ are also removed from the ground set of these contracted matroids. Recall that the contraction of a matroid $\mathcal{M}_i = (\mathcal{N}_i, \mathcal{I}_i)$ to a set $A$ is defined by a matroid $\mathcal{M}'_i = (\mathcal{N} \setminus A, \mathcal{I}'_i)$ such that $\mathcal{I}'_i = \{S \subseteq \mathcal{N} \setminus A : S \cup A \in \mathcal{I}_i\}$. To obtain a solution $S_{a',a''}$, we run Algorithm 1 on the reduced instance. Finally, we return the best solution of $S_{a',a''} \cup \{a', a''\}$ over all feasible pairs $\{a', a''\}$. Here, by construction, we guarantee that all the solutions $S_{a',a''} \cup \{a', a''\}$ are feasible in the original set of constraints. If there is no feasible pair of elements, for the final solution we just return the most valuable feasible singleton element. The details of our algorithm (called BARRIER-GREEDY++) are described in Algorithm 2 in Appendix B.1. Theorem 4.2 guarantees the performance of BARRIER-GREEDY++. The proof for Theorem 4.2 is given in Appendix B.2.

**Theorem 4.2.** BARRIER-GREEDY++ *provides a* $(k+1+\varepsilon)$*-approximation for maximizing a monotone submodular function subject to the intersection of* $k$ *matroids and* $\ell$ *knapsack constraints (for* $\ell \leq k$*). It performs* $O(\frac{n^3 r + n^2 r^3}{\varepsilon} \log r \log 1/\varepsilon)$ *value oracle calls and* $O(\frac{n^3 k r^2}{\varepsilon} \log r \log 1/\varepsilon)$ *membership oracle calls, where* $r$ *is the maximum cardinality of a feasible solution.*

## 5 Experimental Results

In this section, we compare the performance of our proposed algorithms with several baselines. Our first baseline is the vanilla Greedy algorithm. It starts with an empty set $S = \varnothing$ and keeps adding elements one by one greedily (according to their marginal gain) while the $k$-system and $\ell$-knapsack constraints are both satisfied. Our second baseline, Density Greedy, starts with an empty set $S = \varnothing$ and keeps adding elements greedily by the ratio of their marginal gain to the total knapsack cost of each element (i.e., according to ratio $f(a|S)/\gamma_a$ for $e \in \mathcal{N}$) while the $k$-system and $\ell$-knapsack constraints are satisfied. We also consider the state-of-the-art algorithm (called Fast) for maximizing monotone and submodular functions under a constraints and $\ell$ knapsack constraints [1]. This algorithm is a greedy-like algorithm with respect to marginal values, while it discards all elements with a density below some threshold. This thresholding idea guarantees that the solution does not exceed the knapsack constraints without reaching a high enough utility. Furthermore, we compare our proposed algorithms with FANTOM [33] which maximizes a submodular function (not necessarily monotone) subject to intersection of a $k$-system and $\ell$ knapsacks constraints.

In Section 5.1 and Appendix E.1, we compare algorithms on two tasks of video summarization and vertex cover over real-world networks subject to the intersection of matroids and a single knapsack

constraint. Then, in Section 5.2 and Appendices E.2 and E.3, we evaluate algorithms on the Yelp location summarization, Twitter text summarization and movie recommendation applications, respectively, subject to a set system (intersection of matroids for Yelp and Twitter, and $k$-matchoid for movie recommendation) and multiple knapsack constraints. The corresponding constraints are explained independently for each specific application. We set $\varepsilon$ to $0.1$ in all experiments.

In our evaluations, we compare the algorithms based on two criteria: objective value and number of calls to the value oracle. Our experimental evaluations demonstrate the following facts:

- The objective values of the BARRIER-GREEDY algorithm (and BARRIER-HEURISTIC for experiments with more than one knapsack) consistently outperform the baseline algorithms.
- The computational complexities of both our proposed algorithms and Fast are quite lower than FANTOM. In addition, while the Fast algorithm provides a better computational guarantee, we observe that for several applications our algorithm exhibits a better performance (in terms of value oracle calls) than Fast (see Figs. 2d, 3c, 3d, 4d and 5c).

We should point out that although our algorithms might not theoretically be suitable for large scale problems (e.g., $\tilde{O}(nr + r^3)$ could be computationally prohibitive for very large values of $n$ and $r$), both BARRIER-GREEDY and BARRIER-HEURISTIC algorithms are fast in practice (they are comparable with FAST) and this makes them applicable in practical scenarios.

## 5.1 Video Summarizing Application

Video summarization, as a key step for faster browsing and efficient indexing of large video collections, plays a crucial role in many learning procedures. As our first application, we aim to summarize a collection of videos from VSUMM dataset [8].[3] Our objective is to select a subset of frames to maximize a utility function $f(S)$ (which represents the diversity of frames). We set limits for the maximum number of allowed frames from each video (referred to as $m_i$), where we consider the same value of $m_i$ for all five videos. We also want to bound the total entropy of the selection as a proxy for the storage size of the selected summary.

To extract features from frames of each video, we apply a pre-trained ResNet-18 model [20]. Then given a set of frames, we define the matrix $M$ such that $M_{ij} = e^{-\lambda \cdot \text{dist}(x_i, x_j)}$, where $\text{dist}(x_i, x_j)$ denotes the Euclidean distance between the feature vectors of $i$-th and $j$-th frames. Matrix $M$ implicitly represents a similarity matrix among different frames of a video. The utility of a set $S \subseteq \mathcal{N}$ is defined as a non-negative monotone submodular objective $f(S) = \log \det(\mathbf{I} + \alpha M_S)$, where $\mathbf{I}$ is the identity matrix, $\alpha > 0$ and $M_S$ is the principal sub-matrix of $M$ indexed by $S$ [21]. Informally, this function is meant to measure the diversity of the vectors in $S$. A knapsack constraint $c$ captures the entropy of each frame. More specifically, for a frame $u$ we define $c(u) = \text{H}(u)/20$.

In Figs. 1a and 1c, we set the maximum number of allowed frames from each video to $m_i = 10$ and compare algorithms for varying values of knapsack budget. We observe that BARRIER-GREEDY returns solutions with a higher utility (up to 50% more than the second-best algorithm), and the running time of the Fast algorithm is lower than our algorithm. This experiment showcases the that BARRIER-GREEDY effectively trades off some amount of computational complexity to increase the objective values by a huge margin. In Figs. 1b and 1d, we evaluate the performance of algorithms based on the maximum number of allowed frames from each video. While the utility of BARRIER-GREEDY exceeds the other baseline algorithms, its computational complexity follows the same behavior as Fig. 1c. Another important observation is that both Greedy and Density Greedy do not have consistent performance across different applications. For example, while in the experiments of Fig. 3a in Appendix E.1 the Greedy algorithm returns solutions with much higher utilities than Density Greedy, in Fig. 1a, the performance of Density Greedy is even slightly better than FANTOM and Fast for the video summarization task. By increasing the value of $m_i$ the computational complexity of BARRIER-GREEDY increases almost linearly (see Fig. 1d). Our theoretical result confirms this observation as the maximum cardinality of a feasible solution $r$ increases linearly by $m_i$ and the value oracle complexity of BARRIER-GREEDY is proportional to $\tilde{O}(r)$.[4] In our extended experiments, we observed an almost similar behavior for all values of $\lambda \in \{0.1, 0.5, 1.0, 10\}$.

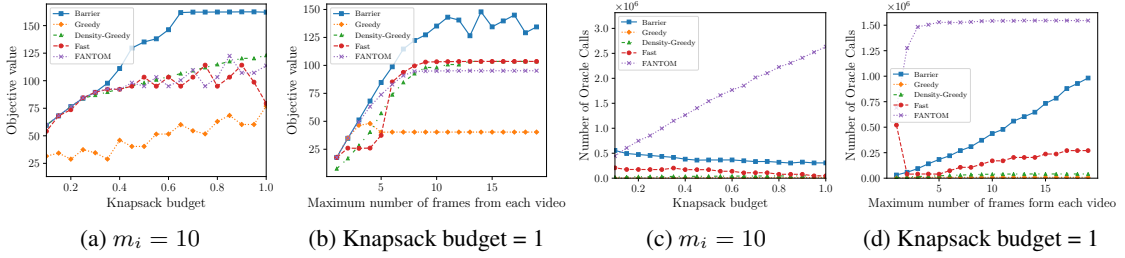

(a) $m_i = 10$      (b) Knapsack budget = 1      (c) $m_i = 10$      (d) Knapsack budget = 1

Figure 1: We summarize a collection of five different videos. (a) and (c) compare algorithms for varying knapsack budgets. (b) and (d) compare algorithms by changing the limit for the maximum number of allowed frames from each video. We also set $\lambda = 1.0$.

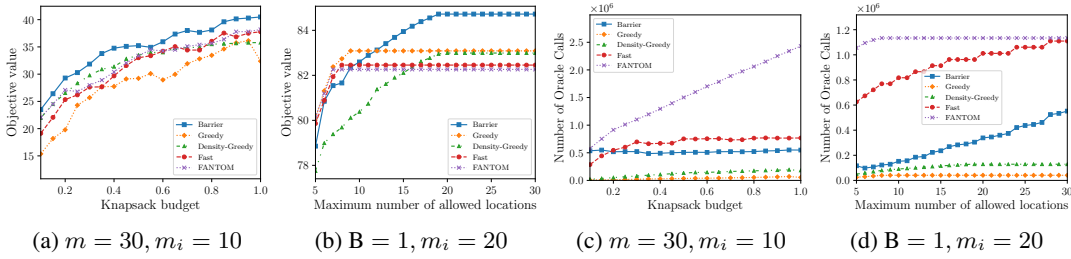

(a) $m = 30, m_i = 10$     (b) B = 1, $m_i = 20$     (c) $m = 30, m_i = 10$     (d) B = 1, $m_i = 20$

Figure 2: Yelp location summarization: A feasible solution satisfies seven different uniform matroids and three knapsack constraints. In $(a)$ and $(c)$, we set $\lambda = 1$. In $(b)$ and $(d)$, we set $\lambda = 1/10$.

## 5.2 More than One Knapsack

In Appendix D, we developed a heuristic algorithm called BARRIER-HEURISTIC to improve the practical performance for cases with multiple knapsacks. Next, we report the result of this algorithm. We consider Yelp location summarization application, where we have access to thousands of business locations with several attributes. We aim to find a representative summary of the locations from the following cities: Charlotte, Edinburgh, Las Vegas, Madison, Phoenix, and Pittsburgh. In these experiments, we use the Yelp Academic dataset [47] which is a subset of Yelp's reviews, business descriptions and user data [48]. For feature extraction, we used the description of each business location and reviews. The features contain information regarding many attributes including having vegetarian menus, existing delivery options and the possibility of outdoor seating.[5]

Suppose we want to select, out of a ground set $\mathcal{N} = \{1, \ldots, n\}$, a subset of locations that provides a good representation of all the existing business locations. The quality of each subset of locations is evaluated by a facility location function which we explain next. A facility at location $i$ is a representative of location $j$ with a similarity value $M_{i,j}$, where $M \in \mathbb{R}^{n \times n}$. For calculating the similarities, similar to the method described in Section 5.1, we use $M_{ij} = e^{-\lambda \cdot \text{dist}(v_i, v_j)}$, where $v_i$ and $v_j$ are extracted feature vectors for locations $i$ and $j$. For a selected set $S$, if each location $i \in \mathcal{N}$ is represented by a location from set $S$ with the highest similarity, the total utility provided by $S$ is modeled by the following monotone and submodular set function [13, 26]: $f(S) = \frac{1}{n} \sum_{i=1}^{n} \max_{j \in S} M_{i,j}$.

For this experiment, we impose a combination of several constraints: (i) there is a limit $m$ on the total size of summary, (ii) the maximum number of locations from each city is $m_i$, and (iii) three knapsacks $c_1, c_2$, and $c_3$ where $c_i(j) = \text{distance}(j, \text{POI}_i)$ is the distance of location $j$ to a point of interest in the corresponding city of $j$. For POIs we consider down-town, an international airport and a national museum in each one of the six cities. One unit of budget is equivalent to 100km, which means the sum of distances of every set of feasible locations to the point of interests (i.e., down-towns, airports or museums) is at most 100km if we set knapsack budget to one.

In Figs. 2a and 2c, we evaluate the performance of algorithms for a varying knapsack budget. We set maximum cardinality of a feasible set to $m = 30$, the maximum number of allowed locations from each city to $m_i = 10$ and $\lambda$ to 1.0. These figures demonstrate that BARRIER-HEURISTIC has the best performance in terms of objective value and outperforms the Fast algorithm with respect to computational complexity. In the second set of experiments, in Figs. 2b and 2d, we compare algorithms based on different upper limits on the total number of allowed locations, where we set the knapsack budgets to one, $m_i$ to 20, and $\lambda$ to 0.1. From our experiments, it is clear that BARRIER-HEURISTIC outperforms Fast, FANTOM and the other baseline algorithms by a huge margin in this setting.

## 6    Conclusion

In this paper, we introduced a novel technique for constrained submodular maximization by borrowing the idea of barrier functions from continuous optimization domain. By using this new technique, we proposed two algorithms for maximizing a monotone and submodular function subject to the intersection of a $k$-matchoid and $\ell$ knapsack constraints. The first algorithm, BARRIER-GREEDY, obtains a $2(k + 1 + \varepsilon)$-approximation ratio and runs in $\tilde{O}(nr + r^3)$ time, where $r$ is the maximum cardinality of a feasible solution. The second algorithm, BARRIER-GREEDY++, improves the approximation factor to $(k + 1 + \varepsilon)$ by increasing the time complexity to $\tilde{O}(n^3 r + n^2 r^3)$. We hope that our proposed method devise new algorithmic tools for constrained submodular optimization that could scale to many previously intractable problem instances. We also extensively evaluated the performance of our proposed algorithms over several real-world applications.

## Broader Impact

The problems studied in this work have deep and far-reaching applications, as scalable data summarization methods play a central role in nearly every scientific and industrial venture in todays information age. Submodular techniques (with applications to human brain parcellation) has the potential to dramatically improve healthcare by reducing risk in delicate medical procedures. Moreover, as machine learning systems are ubiquitously deployed, ensuring fairness and counteracting implicit/historical bias become serious societal considerations. For this reason, a good algorithm that encourages diversity or provides a representative summary could be beneficial to move towards a more fair and just society. On the other hand, an algorithm that fails to summarize the data properly could potentially strengthen the existing historical biases.

## Acknowledgments and Disclosure of Funding

Amin Karbasi is partially supported by NSF (IIS-1845032), ONR (N00014-19-1-2406), AFOSR (FA9550-18-1-0160), and TATA Sons Private Limited.

## Footnotes

[1]Note that $r$ could be in the order of $n$ in the worst case.

[2]In Appendix D, we propose a version of our algorithm that is more aggressive towards approaching the boundaries of knapsack constraints.

[3]Available for download from: `https://sites.google.com/site/vsummsite/`

[4]Note that in the value oracle complexity of Theorem 4.1, the term $nr$ dominates the term $r^3$ for moderate values of $r$. The linearity argument we observe in our experiments is not valid anymore for large values of $r$.

[5]Script is provided at `https://github.com/vc1492a/Yelp-Challenge-Dataset`.

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
