[Supplementary Material]

# A Proof of Theorem 4.1

In this appendix, we provide the Proof of Theorem 4.1 which is omitted from the main text due to the space limitation. We should point out that the choice of our potential function works best for a combination of $k$ matroids and $\ell = k$ knapsacks. When the number of matroid and knapsack constraints is not equal, we can always add redundant constraints so that $k$ is the maximum of the two numbers. For this reason, in the rest of this proof, we assume $\ell = k$.

For this proof, we first show that removing elements $a \in S$ with $\delta_{S,a} \leq 0$ could only decrease the potential function $\phi(S)$.

**Lemma A.1.** *Suppose that $S$ is a current solution such that $\gamma(S) < 1$ and $a \in S$ is such that $\delta_{S,a} \leq 0$. Then, we always have $\phi(S - a) \leq \phi(S)$ with $\gamma(S - a) < 1$.*

*Proof.* First note that by removing an element, the total cost of knapsacks can only decrease, so we still have $\gamma(S') < 1$, as cost of elements is non-negative in all knapsacks. Consider the change in the potential function:

$$\phi(S') - \phi(S) = \frac{\Omega - (k+1)f(S')}{1 - \gamma(S')} - \frac{\Omega - (k+1)f(S)}{1 - \gamma(S)} \qquad \text{(From Eq. (2))}$$

$$= \frac{((\Omega - (k+1)f(S'))(1 - \gamma(S)) - (\Omega - (k+1)f(S))(1 - \gamma(S')))}{(1 - \gamma(S))(1 - \gamma(S'))} \qquad (5)$$

By submodularity, we have $f(S') = f(S - a) \geq f(S) - w_{S,a}$, as for $a \in S$, we have $w_{S,a} = f(S \cap [a]) - f(S \cap [a-1])$. Also, from the linearity of knapsack costs, we have $\gamma(S') = \gamma(S - a) = \gamma(S) - \gamma_a$. Therefore, by applying $f(S') \geq f(S) - w_{S,a}$ and $\gamma(S') = \gamma(S) - \gamma_a$ to the right side of Eq. (5), we get:

$$\phi(S') - \phi(S) \leq \frac{((\Omega - (k+1)f(S) + (k+1)w_{S,a})(1 - \gamma(S)) - (\Omega - (k+1)f(S))(1 - \gamma(S) + \gamma_a))}{(1 - \gamma(S))(1 - \gamma(S'))}$$

$$= \frac{((k+1)w_{S,a}(1 - \gamma(S)) - (\Omega - (k+1)f(S))\gamma_a)}{(1 - \gamma(S))(1 - \gamma(S'))}$$

$$= \frac{\delta_{S,a}}{(1 - \gamma(S))(1 - \gamma(S'))} \leq 0 \ . \qquad (\delta_{S,a} \leq 0 \text{ and } \gamma(S) \leq 1)$$

$\square$

After removing all elements $a \in S$ with $\delta_{S,a} \leq 0$, we obtain a new solution $S$ such that $\delta_a > 0$ for all $a \in S$. In the next step, we require to include a new element in order to decrease the potential function the most. The following lemma provides an algorithmic procedure to achieve this goal. Recall that we denote the $i$-th matroid constraint by $\mathcal{M}_i = (N, \mathcal{I}_i)$.

**Lemma A.2.** *Assume $\mathtt{OPT} = f(S^*) \geq \Omega$, and $S$ is the current solution such that $S \in \cap_{i=1}^k \mathcal{I}_i$, $f(S) < \frac{1}{k+1}\Omega$, and $\gamma(S) < 1$. Assume that for each $a \in S$, $\delta_{S,a} > 0$. Given $b \notin S$, let $J_b = \{i \in [k] : S + b \notin \mathcal{I}_i\}$, and $a_i(b) = argmin\{\delta_{S,a} : a \in S \text{ and } S - a + b \in \mathcal{I}_i\}$ for each $i \in J_b$. Then there is $b \notin S$ such that*

$$\delta_{S,b} - \sum_{i \in J_b} \delta_{S,a_i(b)} \geq \frac{1}{|S^*|}(1 - \gamma(S))(\Omega - (k+1)f(S)) \ .$$

*Proof.* To prove this lemma, we first state the following well-known result for exchange properties of matroids.

**Lemma A.3** ([37], Corollary 39.12a). *Let $\mathcal{M} = (\mathcal{N}, \mathcal{I})$ be a matroid and let $S, T \in \mathcal{I}$ with $|S| = |T|$. Then there is a perfect matching $\pi$ between $S \setminus T$ and $T \setminus S$ such that for every $e \in S \setminus T$, the set $(S \setminus \{e\}) \cup \{\pi(e)\}$ is an independent set.*

Let $S^*$ be an optimal solution with $\mathtt{OPT} = f(S^*) \geq \Omega$. Let us assume that $\widetilde{S}_i, \widetilde{S^*}_i$ are bases of $\mathcal{M}_i$ containing $S$ and $S^*$, respectively. By Lemma A.3, there is a perfect matching $M_i$ between $\widetilde{S}_i \setminus \widetilde{S^*}_i$ and $\widetilde{S^*}_i \setminus \widetilde{S}_i$ such that for any $e \in M_i$, $\widetilde{S}_i \Delta e \in \mathcal{I}_i$. For each $b \in S^*$ and $i \in J_b$ (defined as above,

$J_b$ denotes the matroids in which we cannot add $b$ without removing something from $S$), let $m_i(b)$ denote the endpoint in $S$ of the edge matching $b$ in $M_i$. This means that $S - m_i(b) + b \in \mathcal{I}_i$.

Since for each $i \in J_b$, we pick $a_i(b)$ to be an element of $S$ minimizing $\delta_{S,a}$ subject to the condition $S - a + b \in \mathcal{I}_i$, and $m_i(b)$ is a possible candidate for $a_i$, we have $\delta_{S,a_i(b)} \leq \delta_{S,m_i(b)}$. Consequently, it is sufficient to bound $\delta_{S,b} - \sum_{i \in J_b} \delta_{S,m_i(b)}$ to prove the lemma.

Since each $a \in S$ is matched exactly once in each matching $M_i$, we obtain that each $a \in S$ appears as $m_i(b)$ at most $k$ times for different $i \in [k]$ and $b \in S^*$. Note that it could appear less than $k$ times due to the fact that it might be matched to elements in $\widetilde{S^*}_i \setminus S^*$. Let us define $T_b$ for each $b \in S^*$ to contain $\{m_i(b) : i \in J_b\}$ plus some arbitrary additional elements of $S$, so that each element of $S$ appears in *exactly* $k$ sets $T_b$. Since $\delta_{S,a} > 0$ for all $a \in S$, we have

$$\delta_{S,b} - \sum_{a \in T_b} \delta_{S,a} \leq \delta_{S,b} - \sum_{i \in J_b} \delta_{S,m_i(b)} \leq \delta_{S,b} - \sum_{i \in J_b} \delta_{S,a_i(b)} \ .$$

Hence it is sufficient to prove that $\delta_{S,b} - \sum_{a \in T_b} \delta_{S,a} \geq \frac{1}{|S^*|}(1 - \gamma(S))(\Omega - (k+1)f(S))$ for some $b \in S^*$. Let us choose a random $b \in S^*$ and compute the expectation $\mathbf{E}[\delta_{S,b} - \sum_{a \in T_b} \delta_{S,a}]$. First, since each element of $S^*$ is chosen with probability $\frac{1}{|S^*|}$, we obtain

$$\mathbf{E}[w_{S,b}] = \frac{\sum_{b \in S^*} w_{S,b}}{|S^*|} \geq \frac{\sum_{b \in S^*} f(b \mid S)}{|S^*|} \geq \frac{f(S^* \mid S)}{|S^*|} =\geq \frac{\Omega - f(S)}{|S^*|} \ ,$$

by monotonicity and submodularity of function $f$. Similarly, since $S^*$ is a feasible solution, we have

$$\mathbf{E}[\gamma_b] = \frac{1}{|S^*|} \sum_{b \in S^*} \gamma_b \leq \frac{k}{|S^*|} \ .$$

Concerning the contribution of the items in $T_b$, we obtain,

$$\mathbf{E}\Big[\sum_{a \in T_b} w_{S,a}\Big] = \frac{1}{|S^*|} \sum_{b \in S^*} \sum_{a \in T_b} w_{S,a} = \frac{k}{|S^*|} \sum_{a \in S} w_{S,a} = \frac{k}{|S^*|} f(S) \ ,$$

using the fact that each $a \in S$ appears in exactly $k$ sets $T_b$. Similarly,

$$\mathbf{E}\Big[\sum_{a \in T_b} \gamma_a\Big] = \frac{1}{|S^*|} \sum_{b \in S^*} \sum_{a \in T_b} \gamma_a = \frac{k}{|S^*|} \gamma(S) \ .$$

All together, we obtain

$$\mathbf{E}[\delta_{S,b} - \sum_{a \in T_b} \delta_{S,a}] = \mathbf{E}\Bigg[(k+1) \cdot (1 - \gamma(S)) \cdot (w_{S,b} - \sum_{a \in T_b} w_{S,a})$$

$$- (\Omega - (k+1) \cdot f(S)) \cdot (\gamma_b - \sum_{a \in T_b} \gamma_{a_i})\Bigg]$$

$$\geq \frac{k+1}{|S^*|} \cdot (1 - \gamma(S)) \cdot (\Omega - f(S) - k \cdot f(S)) - \frac{1}{|S^*|} \cdot (\Omega - (k+1) \cdot f(S)) \cdot (k - k \cdot \gamma(S))$$

$$= \frac{1}{|S^*|} \cdot (1 - \gamma(S)) \cdot (\Omega - (k+1) \cdot f(S)) \ .$$

Since the expectation is at least $\frac{1}{|S^*|} \cdot (1 - \gamma(S)) \cdot (\Omega - (k+1) \cdot f(S))$, there must exist an element $b \in S^*$ for which the expression is at least the same amount, which proves the lemma. $\qquad \square$

Now, we bound the maximum required number of iterations to converge to a solution whose value is sufficiently high. Let $r = |S^*|$ and $\mathtt{OPT} = f(S^*)$ for the optimal solution $S^*$. In Algorithm 1, we start from $S = \varnothing$ and repeat the following: As long as $\delta_{S,a} < 0$ for some $a \in S$, we remove $a$ from $S$. If there is no such $a \in S$, we find $b \notin S$ such that $\delta_{S,b} - \sum_{i \in J_b} \delta_{S,a_i(b)} \geq \frac{1}{|S^*|}(1 - \gamma(S))(\Omega - (k+1)f(S))$ (see Lemma A.2); we include element $b$ in $S$ and remove set $J_b$ from $S$.

**Lemma A.4.** BARRIER-GREEDY, *after at most* $r \log(1/\varepsilon)$ *iterations, returns a set $S$ such that* $f(S) > \frac{1-\varepsilon}{k+1}\Omega$. *Furthermore, at least one of the two sets $S$ or $S - b$ is feasible, where $b$ is the last element added to $S$.*

*Proof.* At the beginning of the process, we have $\phi(\varnothing) = \Omega$. Our goal is to show that $\phi(S)$ decreases sufficiently fast, while we keep the invariant $0 \leq \gamma(S) < 1$.

We know that, from the result of Lemma A.1, removing elements $a \in S$ with $\delta_{S,a} \leq 0$ can only decrease the value of $\phi(S)$. We ignore the possible gain from these steps. When we include a new element $b$ and remove $\{a_i(b) : i \in J_b\}$ from $S$, we get from Lemma A.2:

$$\delta_{S,b} - \sum_{i \in J_b} \delta_{S,a_i(b)} \geq \frac{1}{|S^*|} \cdot (1 - \gamma(S)) \cdot (\Omega - (k+1) \cdot f(S)) \ .$$

Next, let us relate this to the change in $\phi(S)$. We denote the modified set by $S' = (S+b) \setminus \{a_i(b) : i \in J_b\}$. First, by submodularity and the definition of $w_{S,a}$, we know that

$$f(S') \geq f(S) + w_{S,b} - \sum_{i \in J_b} w_{S,a_i(b)} \ .$$

We also have

$$\gamma(S') = \gamma(S) + \gamma_b - \sum_{i \in J_b} \gamma_{a_i(b)} \ .$$

First, let us consider what happens when $\gamma(S') \geq 1$. This means that $\gamma_b - \sum_{i \in J_b} \gamma_{a_i(b)} \geq 1 - \gamma(S)$. Since we know that $\delta_{S,b} - \sum_{i \in J_b} \delta_{S,a_i(b)} \geq 0$, this means (by the definitions of $\delta_{S,b}$ and $\delta_{S,a_i(b)}$) that

$$(k+1) \cdot (w_{S,b} - \sum_{i \in J_b} w_{S,a_i(b)}) \geq \Omega - (k+1) \cdot f(S) \ .$$

In other words, $f(S') \geq f(S) + w_{S,b} - \sum_{i \in J_b} w_{S,a_i} \geq \frac{1}{k+1}\Omega$. Note that $S'$ might be infeasible, but $S' - b$ is feasible (since $S$ was feasible), so in this case we are done.

In the following, we assume that $\gamma(S') < 1$. Then the potential change is

$\phi(S') - \phi(S)$

$$\leq \frac{\left(\Omega - (k+1)\left(f(S) + w_{S,b} - \sum_{i \in J_b} w_{S,a_i(b)}\right)\right)(1 - \gamma(S))}{(1 - \gamma(S))(1 - \gamma(S'))}$$

$$- \frac{\left(\Omega - (k+1)f(S)\right)\left(1 - \gamma(S) - \gamma_b + \sum_{i \in J_b} \gamma_{a_i(b)}\right)}{(1 - \gamma(S))(1 - \gamma(S'))}$$

$$= \frac{\left((k+1)(-w_{S,b} + \sum_{i \in J_b} w_{S,a_i(b)})(1 - \gamma(S)) - (\Omega - (k+1)f(S))(-\gamma_b + \sum_{i \in J_b} \gamma_{a_i(b)})\right)}{(1 - \gamma(S))(1 - \gamma(S'))}$$

$$= \frac{(-\delta_{S,b} + \sum_{i \in J_b} \delta_{S,a_i(b)})}{(1 - \gamma(S))(1 - \gamma(S'))} \leq -\frac{1}{|S^*|}\frac{\Omega - (k+1)f(S)}{1 - \gamma(S')}$$

$$= -\frac{1}{r}\frac{1 - \gamma(S)}{1 - \gamma(S')}\phi(S) \ ,$$

using Lemma A.2. We infer that

$$\phi(S') \leq \left(1 - \frac{1}{r} \cdot \frac{1 - \gamma(S)}{1 - \gamma(S')}\right)\phi(S) \ .$$

By induction, if we denote by $S_t$ the solution after $t$ iterations,

$$\phi(S_t) \leq \prod_{i=1}^{t}\left(1 - \frac{1}{r} \cdot \frac{1 - \gamma(S_{i-1})}{1 - \gamma(S_i)}\right)\phi(S_0) \leq e^{-\frac{1}{r}\sum_{i=1}^{t}\frac{1 - \gamma(S_{i-1})}{1 - \gamma(S_i)}}\phi(S_0) \ .$$

Here, we use the arithmetic-geometric-mean inequality:

$$\frac{1}{t}\sum_{i=1}^{t}\frac{1 - \gamma(S_{i-1})}{1 - \gamma(S_i)} \geq \left(\prod_{i=1}^{t}\frac{1 - \gamma(S_{i-1})}{1 - \gamma(S_i)}\right)^{1/t} = \left(\frac{1 - \gamma(S_0)}{1 - \gamma(S_t)}\right)^{1/t} \geq 1 \ .$$

Therefore, we can upper bound the potential function at the iteration $t$:

$$\phi(S_t) \leq e^{-\frac{t}{r} \cdot \frac{1}{t} \sum_{i=1}^{t} \frac{1-\gamma(S_{i-1})}{1-\gamma(S_i)}} \phi(S_0) \leq e^{-\frac{t}{r}} \phi(S_0) = e^{-\frac{t}{r}} \Omega \ .$$

For $t = r \log \frac{1}{\varepsilon}$, we obtain $\phi(S_t) = \frac{\Omega - (k+1)f(S_t)}{1-\gamma(S_t)} \leq \varepsilon \Omega$ (and $0 \leq \gamma(S_t) < 1$), which implies $f(S_t) \geq \frac{1-\varepsilon}{k+1}\Omega$. $\qquad \square$

Now, we have all the required materials to prove Theorem 4.1.

**Proof of Theorem 4.1** The for loop for estimating OPT is repeated $\frac{1}{\varepsilon} \log r$ times. Consider the value of $\Omega$ such that $(1-\varepsilon)OPT \leq \Omega \leq OPT$. We perform the local search procedure: In each iteration, we check all possible candidates $b \in \mathcal{N} \setminus S$ and find the best swap $a_i$ for each matroid $\mathcal{M}_i$ where a swap is needed (the set of indices $J_b$). This requires checking the membership oracles for $\mathcal{M}_i$ and the values $\delta_{a_i}$ for each potential swap. This takes $O(rkn)$ calls to the membership oracle. Finally, we choose the elements $b \notin S$ and $a_i \in S$ so that $\delta_{S,b} - \sum_{i \in J_b} \delta_{S,a_i}$ is maximized. Note that, in each iteration of the while loop, we need to compute $\delta_{S,a}$ for all elements $a \in \mathcal{N}$ only once and store them; then we can use these precomputed values to find the best candidate $b$. Therefore, in each iteration, algorithm needs to make $O(n)$ calls to the evaluation oracle to find $b$. Also, Line 8 of 1 needs $O(r^2)$ calls to the evaluation oracle. To sum-up, each iteration of the while loop (Line 5) requires $O(n+r^2)$ calls to the evaluation oracle and $O(rkn)$ calls to the membership oracle. Due to Lemma A.2, the best swap satisfies $\delta_{S,b} - \sum_{i \in J_b} \delta_{S,a_i} \geq \frac{1}{r}(1-\gamma(S))(\Omega - (k+1)f(S))$. Following this swap, we need to recompute the values of $\delta_{S,a}$ for $a \in S$ and remove all elements with $\delta_{S,a} \leq 0$. Considering Lemma A.4, it is sufficient to prove that we terminate within $O(r \log \frac{1}{\varepsilon})$ iterations of the local search procedure. Therefore, the algorithm terminates with $O(\frac{nr+r^3}{\varepsilon} \log r \log \frac{1}{\varepsilon})$ calls to the evaluation oracle and $O(\frac{nkr^2}{\varepsilon} \log r \log \frac{1}{\varepsilon})$ calls to the membership oracle. In the end, we have a set $S$ such that $f(S) \geq \frac{1-\varepsilon}{k+1}\Omega$ (as the result of Lemma A.4). It is possible that $S$ is infeasible, but both $S - b$ and $b$ are feasible (where $b$ is the last-added element), and by submodularity one of them has an objective value of at least $\frac{1-\varepsilon}{2k+2}\Omega$.

# B The BARRIER-GREEDY++ Algorithm

## B.1 The Pseudocode for BARRIER-GREEDY++

In this section, we detail BARRIER-GREEDY++ in Algorithm 2.

---

**Algorithm 2** BARRIER-GREEDY++

---

**Input:** $f : 2^{\mathcal{N}} \to \mathbb{R}_+$, membership oracles for $k$ matroids $\mathcal{M}_1 = (\mathcal{N}, \mathcal{I}_1), \dots, \mathcal{M}_k = (\mathcal{N}, \mathcal{I}_k)$, and $\ell$ knapsack-cost functions $c_i : \mathcal{N} \to [0, 1]$.
**Output:** A set $S \subseteq \mathcal{N}$ satisfying $S \in \bigcap_{i=1}^{k} \mathcal{I}_i$ and $c_i(S) \leq 1 \ \forall i$.
1: **for** each feasible pair of elements $\{a', a''\}$ **do**
2: $\quad g(S) \triangleq f(S \cup \{a', a''\}) - f(\{a', a''\})$.
3: $\quad$ Decrease the knapsack capacities by $c_{i,a'} + c_{i,a''}$.
4: $\quad \mathcal{N}' \leftarrow \mathcal{N} \setminus (\{a', a''\} \cup \{a \mid g(a) > \frac{1}{2}f(\{a', a''\}))$ and contracts all matroids $\mathcal{M}_i(\mathcal{N}_i, \mathcal{I}_i)$ by set $\{a', a''\}$.
5: $\quad$ Run Algorithm 1 on the reduced instance $g : 2^{\mathcal{N}'} \to \mathbb{R}_+$, to obtain a solution $S_{a',a''}$.
6: **return** the best of $S_{a',a''} \cup \{a', a''\}$ over all feasible pairs $\{a', a''\}$ (If there is no feasible pair of elements, just return the most valuable singleton).

---

## B.2 Proof of Theorem 4.2

*Proof.* Since we enumerate over $O(n^2)$ pairs of elements, the running time is $O(n^2)$ times the running time of Algorithm 1.

Consider an optimal solution $S^*$ and a greedy ordering of its elements with respect to $f$. Also, consider the run of the algorithm, when $a', a''$ are the first two elements of $S^*$ in the greedy ordering.

Note that if all optimal solutions have only one element, it means there is no feasible pair, due to the monotonicity of $f$. In this case, we just return the best singleton, which is optimal. All elements of $S^*$ following $a', a''$ in the greedy ordering have a marginal value of at most $\frac{1}{2}f(\{a', a''\})$, by the greedy choice of $a', a''$. Therefore, these elements are still present in the reduced instance. Furthermore, since $S^* \setminus \{a', a''\}$ is a feasible solution in the reduced instance, Algorithm 1 always finds a solution: if the produced set $S$ by Algorithm 1 is feasible, then the solution is returned at Line 10 of that algorithm with a guarantee:

$$g(S) \geq \frac{1 - \varepsilon}{k + 1} g(S^* \setminus \{a', a''\}) = \frac{1 - \varepsilon}{k + 1}(OPT - f(\{a', a''\})) \ .$$

However, the set $S$ could be potentially infeasible and the solution then is returned at Line 12 of Algorithm 1. In this case, we know that $S - b$ is feasible in the reduced instance where $b$ is the last-added element, and hence $S - b + a' + a''$ is feasible in the original instance. Also, $g(b) \leq \frac{1}{2}f(\{a', a''\})$, otherwise $b$ would not be present in the reduced instance. By submodularity, the value of $S - b + a' + a''$ is at least

$$
\begin{aligned}
f(S - b + a' + a'') = f(\{a', a''\}) + g(S - b) &\geq f(\{a', a''\}) + g(S) - g(\{b\}) \\
&\geq f(\{a', a''\}) + \frac{1 - \varepsilon}{k + 1}(OPT - f(\{a', a''\})) - \frac{1}{2}f(\{a', a''\}) \\
&\geq \frac{1 - \varepsilon}{k + 1} OPT \ .
\end{aligned}
$$

Since $S + a' + a''$ or $S - b + a' + a''$ is one of the considered solutions, we are done. □

## C  The Generalization to $k$-matchoids

In this section, we show that our algorithms could be extended to $k$-matchoids, a more general class of constraints. To achieve this goal, we need to slightly modify the BARRIER-GREEDY algorithm to make it suitable for the $k$-matchoid constraint. More specifically, for each element $b \in S$, we use EXCHANGECANDIDATE to find a set $U_b \subseteq S$ such that $(S \setminus U_b) + b$ satisfies the $k$-matchoid constraint where exchanges are done with elements with the minimum values of $\delta_{S,a}$. The pseudocode of EXCHANGECANDIDATE is given as Algorithm 3.

---
**Algorithm 3** EXCHANGECANDIDATE $(S, b)$
---
1: Let $U \leftarrow \varnothing$.
2: **for** $i = 1$ **to** $m$ **do**
3:     **if** $(S + b) \cap \mathcal{N}_i \notin \mathcal{I}_i$ **then**
4:         Let $A_i \leftarrow \{a \in S \mid ((S - a + b) \cap \mathcal{N}_i) \in \mathcal{I}_\ell\}$.
5:         Let $a_i \leftarrow \arg\min_{a \in A_\ell} \delta_{S,a}$.
6:         Add $a_i$ to $U$.
7: **return** $U$.
---

In order to guarantee the performance of our proposed algorithms under the $k$-matchoid constraint, we provide the following lemma which is the equivalent of Lemma A.2 for $k$-matchoid.

**Lemma C.1.** *Assume* $\text{OPT} = f(S^*) \geq \Omega$, *and* $S$ *is the current solution that satisfies the $k$-matchoid constraint* $\mathcal{M}(\mathcal{N}, \mathcal{I})$ *with* $f(S) < \frac{1}{k+1}\Omega$, *and* $\gamma(S) < 1$. *Then there is* $b \notin S$ *such that*

$$\delta_{S,b} - \sum_{i \in J_b} \delta_{S,a_i(b)} \geq \frac{1}{|S^*|}(1 - \gamma(S))(\Omega - (k + 1)f(S)) \ .$$

*Proof.* For the sake of simplicity of analysis, we assume that every element $a \in \mathcal{N}$ belongs to exactly $k$ out of the $m$ ground sets $\mathcal{N}_i$ ($i \in [m]$) of the matroids defining $\mathcal{N}$. To make this assumption valid, for every element $a \in \mathcal{N}$ that belongs to the ground sets of only $k' < k$ out of the $m$ matroids, we add $a$ to $k - k'$ additional matroids as an element whose addition to an independent set always keeps the set independent. It is easy to observe that the addition of $a$ to these matroids does not affect the behavior of our Algorithms.

Let us assume that $\widetilde{S}_i, \widetilde{S^*}_i$ are bases of $\mathcal{M}_i$ containing $S \cap \mathcal{N}_i$ and $S^* \cap \mathcal{N}_i$, respectively. By Lemma A.3, there is a perfect matching $\Pi_i$ between $\tilde{S}_i \setminus \widetilde{S^*}_i$ and $\widetilde{S^*}_i \setminus \widetilde{S}_i$ such that for any $e \in \Pi_i$ we have $\widetilde{S}_i \Delta e \in \mathcal{I}_i$. For each $b \in S^*$ and $i \in J_b$ where we define $J_b = \{i \in [m] \mid (S + b) \cap \mathcal{N}_i \notin \mathcal{I}_i\}$, let $\pi_i(b)$ denote the endpoint in $S$ of the edge matching $b$ in $\Pi_i$. This means that $S - \pi_i(b) + b \in \mathcal{I}_i$. Since for each $i \in J_b$, we pick $a_i(b)$ to be an element of $S$ minimizing $\delta_{S,a}$ subject to the condition $S - a + b \in \mathcal{I}_i$, and $\pi_i(b)$ is a possible candidate for $a_i$, we have $\delta_{S,a_i(b)} \leq \delta_{S,\pi_i(b)}$. Consequently, it is sufficient to bound $\delta_{S,b} - \sum_{i \in J_b} \delta_{S,\pi_i(b)}$ to prove the lemma. Since each $a \in S$ is matched at most once in each matching $\Pi_i$, we obtain that each $a \in S$ appears as $\pi_i(b)$ at most $k$ times for different $i \in [m]$ and $b \in S^*$. Note that it could appear less than $k$ times. We can then define $T_b$ for each $b \in S^*$ to contain $\{\pi_i(b) : i \in J_b\}$ plus some arbitrary additional elements of $S$, so that each element of $S$ appears in exactly $k$ sets $T_b$. By providing this exchange property for $k$-matchoids, the rest of the proof is the same as proof of Lemma A.2. $\qquad\square$

From the result of Lemma C.1 and Theorems 4.1 and 4.2, we conclude the following corollaries for maximizing a monotone and submodular function subject to a $k$-matchoid and $\ell$ knapsack constraints.

**Corollary C.2.** BARRIER-GREEDY *(Algorithm 1) provides a $2(k + 1 + \varepsilon)$-approximation for the problem of maximizing a monotone submodular function subject to $k$-matchoid and $\ell$ knapsack constraints (for $\ell \leq k$).*

**Corollary C.3.** BARRIER-GREEDY++ *(Algorithm 2) provides a $(k + 1 + \varepsilon)$-approximation for the problem of maximizing a monotone submodular function subject to $k$-matchoid and $\ell$ knapsack constraints (for $\ell \leq k$).*

# D   The BARRIER-HEURISTIC Algorithm

In Section 4, we proposed BARRIER-GREEDY with the following property: it needs to consider only sets $S$ where the sum of all the $k$ knapsacks is at most 1 for them, i.e., sets $S$ such that $\gamma(S) = \sum_i^k \sum_{a \in S} c_{i,a} \leq 1$. For scenarios with more than one knapsack, while BARRIER-GREEDY theoretically produces a highly competitive objective value, there might be feasible solutions such that they fill the capacity of all knapsacks, i.e., $\gamma(S)$ could be very close to $k$ for them. Unfortunately, our proposed algorithms fail to find these kinds of solutions. In this section inspired by our theoretical results, we design a heuristic algorithm (called BARRIER-HEURISTIC) that overcomes this issue. More specifically, this algorithm is very similar to BARRIER-GREEDY with two slight modifications: (i) Instead of Eq. (4), we use a new formula to calculate the importance of an element $a$ with respect to the potential function:

$$\delta_{S,a} = (k+1)(\lambda - \gamma(S))w_{S,a} - (\Omega - (k+1)f(S))\gamma_a \ , \qquad (6)$$

where $1 \leq \lambda \leq k$. This modification allows us to include sets with $\gamma(S) > 1$ for the outcome of algorithms as $\delta_{S,a}$ could still be non-negative for them. (ii) The BARRIER-GREEDY is designed in a way such that for a solution $S$, we have $\gamma(S) \leq 1$. This fact consequently implies that the set $S$ satisfies all the knapsack constraints; therefore, by the algorithmic design, we can guarantee that knapsacks are not violated. On the other hand, in Eq. (6) for values $\lambda > 1$, set $S$ may violate one or more of the knapsack constraints. For this reason, we need to choose the element $b$ from a set $\mathcal{N}'$ such that for all $b \in \mathcal{N}'$ the set $(S \setminus U_b) + b$ is feasible; and if this set $\mathcal{N}'$ is empty, i.e., there is no such element $b$, we stop the algorithm and return the solution (see Line 7 of Algorithm 4). For the sake of completeness, we provide a detailed description of BARRIER-HEURISTIC in Algorithm 4.

In Appendix D, inspired by our theoretical results, we proposed an algorithm with a better performance in applications with more than one knapsack. Recall we defined a modified formula for $\delta_{S,a}$ in Eq. (6):

$$\delta_{S,a} = (k+1)(\lambda - \gamma(S))w_{S,a} - (\Omega - (k+1)f(S))\gamma_a \ ,$$

where $1 \leq \lambda \leq k$. We explained that this new choice of $\delta_{S,a}$ allows the BARRIER-HEURISTIC algorithm to choose subsets whose their weights fill almost all the $\ell$ knapsacks. BARRIER-HEURISTIC, with the objective of maximizing a monotone and submodular function $f$ subject to the intersection of a $k$-matchoid and $\ell$ knapsack constraints, is given as Algorithm 4.

**Algorithm 4** BARRIER-HEURISTIC

---

**Input:** $f : 2^{\mathcal{N}} \to \mathbb{R}_{\geq 0}$, membership oracles for a $k$-matchoid set system $(\mathcal{N}, \mathcal{I})$, and $\ell$ knapsack-cost functions $c_i : \mathcal{N} \to [0, 1]$.
**Output:** A set $S \subseteq \mathcal{N}$ satisfying $S \in \mathcal{I}$ and $c_i(S) \leq \forall i$.
1: $M \leftarrow \max_{j \in \mathcal{N}} f(\{j\})$
2: $\Lambda \leftarrow \{(1 + \varepsilon)^i \mid M/(1+\varepsilon) \leq (1 + \varepsilon)^i \leq rM\}$ as potential estimates of OPT
3: **for** $\Omega \in \Lambda$ **do**
4:     $S \leftarrow \varnothing$.
5:     **for** Iteraton number from 1 **to** $r \log \frac{1}{\varepsilon}$ **do**
6:         $\mathcal{N}' \leftarrow \{b \in \mathcal{N} \setminus S \mid (S \setminus U_b) + b$ satisfies all knapsacks$\}$, where $U_b \leftarrow$ EXCHANGECANDIDATE$(S, b)$.
7:         **if** $\mathcal{N}' = \varnothing$ **then break**.
8:         $b \leftarrow \operatorname{argmax}_{b \in \mathcal{N}'} \left( \delta_{S,b} - \sum_{a \in U_b} \delta_{S,a} \right)$ for $\delta_{S,a} = (k+1)(\lambda - \gamma(S))w_{S,a} - (\Omega - (k+1)f(S))\gamma_a$.
9:         $S \leftarrow (S \setminus U_b) + b$, where $U_b \leftarrow$ EXCHANGECANDIDATE$(S, b)$.
10:        **while** $\exists a \in S$ such that $\delta_{S,a} \leq 0$ **do** Remove $a$ from $S$.
11:    $S_\Omega \leftarrow S$
12: **return** $\operatorname{argmax}_{\Omega \in \Lambda} f(S_\Omega)$

---

# E   Supplementary Experiments

In Appendix E.1, we evaluate the performance of algorithms on a vertex cover application over real-world networks. In Appendix E.2, we evaluate the performance of algorithms on a Twitter data summarization task. Finally, in Appendix E.3, we consider a movie recommendation application.

## E.1   Vertex Cover

In this experiment, we compare BARRIER-GREEDY with Greedy, Density Greedy, Fast and FANTOM. We define a monotone and submodular function over vertices of a directed real-world graph $G = (V, E)$. Let's $w : V \to \mathbb{R}_{\geq 0}$ denotes a weight function on the vertices of graph $G$. For a given vertex set $S \subseteq V$, assume $N(S)$ is the set of vertices which are pointed to by $S$, i.e., $N(S) \triangleq \{v \in V \mid \exists u \in S$ such that $(u, v) \in E\}$. We define $f : 2^V \to \mathbb{R}_{\geq 0}$ as follows:

$$f(S) = \sum_{u \in N(S) \cup S} w_u \ ,$$

and we assign to each vertex $u$ a weight of one. In this set of experiments, our objective is to maximize function $f$ subject to the constraint that we have an upper limit $m$ on the total number of vertices we choose, as well as an upper limit $m_i$ on the number of vertices from each social communities. For the simplicity of our evaluations, we use a single value for all $m_i$. This constraint is the intersection of a uniform matroid and a partition matroid. To assign vertices to different communities, we use the Louvain method [3][6]. Besides, for each graph, we reduce the total number of communities to five by merging smaller communities. For a knapsack constraint $c$, we set the cost of each vertex $u$ as $c(u) \propto 1 + \max\{0, d(u) - q\}$, where $d(u)$ is the out-degree of node $u$ in graph $G(V, E)$. We normalize the costs such that the average cost of each element is $1/20$, i.e., $\frac{\sum_{u \in V} c(u)}{|V|} = 1/20$. With this normalization, we expect the average size of the largest set which satisfies the knapsack constraint is roughly close to 20. In our experiment, we use real-world graphs from [27] and run the algorithms for varying knapsack budgets. We also set $m = 15, m_i = 6$ and $q = 6$.

In Appendix E.1, we see the evaluations for two graphs: Facebook ego network and EU Email exchange network. From these experiments, it is evident that BARRIER-GREEDY outperforms the other specialized algorithms for this problem in terms of both objective value and computational complexity. We also observe that the performance of Greedy is slightly worse than Fast and FANTOM. We should point out that the running times of Greedy and Density Greedy are the two smallest,

as these two algorithms do not make any adjustments to make them suitable for the constraints of this application and obviously they do not provide any theoretical guarantees.

(a) Facebook network      (b) EU Email      (c) Facebook network      (d) EU Email

Figure 3: Vertex cover over real graphs: We compare algorithms for varying knapsack budges based on objective value and number of calls to the Oracle.

## E.2 Twitter Text Summarization

As of January 2019, six of the top fifty Twitter accounts are dedicated primarily to news reporting. In this application, we want to produce representative summaries for Twitter feeds of several news agencies with the following Twitter accounts (also known as "handles"): @CNNBrk, @BBCSport, @WSJ, @BuzzfeedNews, @nytimes, and @espn. Each of these handles has millions of followers. Naturally, such accounts commonly share the same headlines and it would be very valuable if we could produce a summary of stories that still relays all the important information without repetition.

In this application, we use the Twitter dataset from [21], where the keywords from each tweet are extracted and weighted proportionally to the number of retweets the post received. To capture diversity in a selected set of tweets, similar to the approach of Kazemi et al. [21], we define a monotone and submodular function $f$ defined over a ground set $\mathcal{N}$ of tweets, where we take the square root of the value assigned to each keyword. Each tweet $e \in \mathcal{N}$ consists of a positive value $\text{val}_e$ denoting its number of retweets and a set of $\ell_e$ keywords $\mathcal{W}_e = \{w_{e,1}, \cdots, w_{e,\ell_e}\}$ from the set of all existing keywords $\mathcal{W}$. For a tweet $e$, the score of a word $w \in \mathcal{W}_e$ is defined by $\text{score}(w, e) = \text{val}_e$. If $w \notin \mathcal{W}_e$, we define $\text{score}(w, e) = 0$. The function $f$, for a set $S \subseteq \mathcal{N}$ of tweets, is defined as follows:

$$f(S) = \sum_{w \in \mathcal{W}} \sqrt{\sum_{e \in S} \text{score}(w, e)} \ .$$

A feasible summary should have at most five tweets from each one of the accounts with an upper limit of 15 on the total number of tweets. Again, this constraint is the intersection of a uniform matroid and a partition matroid. Also, it should satisfy existing knapsack constraints. For the first knapsack $c_1$, the cost of each tweet $e$ is weighted proportionally to the difference between the time of $e$ and January 1, 2019, i.e., $c_1(e) \propto |01/01/2019 - \text{T}(e)|$. The goal of this knapsack is to provide a summary that mainly captures the events happened around the beginning of the year 2019. For the second knapsack $c_2$ the cost of tweet $e$ is proportional to the length of each tweet $|\mathcal{W}_e|$ which enables us to provide shorter summaries. Each unit of knapsack budget is equivalent to roughly 10 months for $c_1$ and 26 keywords for $c_2$, respectively.

In Figs. 4a and 4c, we compare algorithms under only one knapsack constraint. Similar to the trends in the previous experiments, we observe that BARRIER-GREEDY provides the best utilities, where its number of Oracle calls is competitive with respect to Fast. In Figs. 4b and 4d, we report the experimental results subject to two knapsacks $c_1$ and $c_2$. We see that BARRIER-HEURISTIC returns the solutions with the highest objective values with a fewer number of calls to the Oracle with respect to Fast. We should emphasize that both Greedy and Density Greedy algorithms, due to their simplicity and lack of theoretical guarantees, have the lowest computational complexities. We also can observe, while the time complexity of FANTOM increases with higher budgets, computational costs remain almost fixed for both our algorithm and FAST. Finally, by comparing the scenarios with one and two knapsacks, it is evident that having more knapsacks reduces objective values and computational complexity. The main reason for this phenomenon is that by imposing more constraints the size of all feasible sets decreases.

(a) One knapsack $c_1$     (b) Two knapsacks $c_1$ and $c_2$     (c) One knapsack $c_1$     (d) Two knapsacks $c_1$ and $c_2$

Figure 4: Twitter text summarization: We compare algorithms based on varying knapsack budget. For knapsacks we have $c_1(e) = |01/01/2019 - \mathrm{T}(e)|$ and $c_2(e) = |\mathcal{W}_e|$.

### E.3 Movielens Recommendation System

In the final application, our objective is to recommend a set of diverse movies to a user. For designing our recommender system, we use ratings from MovieLens dataset [17], and apply the method proposed by Lindgren et al. [29] to extract a set of attributes for each movie. For this experiment, we consider a subset of this dataset which contains 1793 movies from the three genres of Adventure, Animation, and Fantasy. For a ground set of movies $\mathcal{N}$, assume $v_i$ represents the feature vector of the $i$-th movie. Following the same approach we used in Section 5.1, we define a similarity matrix $M$ such that $M_{ij} = e^{-\lambda \cdot \mathrm{dist}(v_i, v_j)}$, where $\mathrm{dist}(v_i, v_j)$ is the euclidean distance between vectors $v_i, v_j \in \mathcal{N}$. The objective of each algorithm is to select a subset of movies that maximizes the following monotone and submodular function: $f(S) = \log \det(\mathbf{I} + \alpha M_S)$, where $\mathbf{I}$ is the identity matrix.

The user specifies an upper limit $m$ on the number of movies for the recommended set, as well as an upper limit $m_i$ on the number of movies from each one of the three genres. This constraint represents a $k$-matchoid independence system with $k = 4$, because a single movie may be identified with multiple genres and the constraint over the genres is not a partition matroid anymore. In addition to this $k$-matchoid constraint, we consider three different knapsacks. For the first knapsack $c_1$, the cost assigned to each movie is proportional to the difference between the maximum possible rating in the iMDB (which is 10) and the rating of the particular movie—here the goal is to pick movies with higher ratings. For the second and third knapsacks $c_2$ and $c_3$, the costs of each movie are proportional to the absolute difference between the release year of the movie and the year 1990 and year 2004. The implicit goal of these knapsack constraints is to pick movies with a release year which is as close as possible to these years. More formally, for a movie $v \in \mathcal{N}$, we have: $c_1(v) = 10 - \mathrm{rating}_v$, $c_2(v) = |1990 - \mathrm{year}_v|$, and $c_3(v) = |2004 - \mathrm{year}_v|$. Here, $\mathrm{rating}_v$ and $\mathrm{year}_v$, respectively, denote the IMDb rating and the release year of movie $v$. We normalize the knapsacks such that the average cost of each movie is $1/10$, i.e., $\frac{\sum_{v \in \mathcal{N}} c_i(V)}{|\mathcal{N}|} = 1/10$. For simplicity, we use a single value $m_i = 20$ for all genres, and we set $\lambda = 0.1$.

In Figs. 5a and 5c, we evaluate the algorithms for varying the maximum number of allowed movies in the recommendation. For the knapsacks, we consider $c_1$ and $c_2$. In this experiment, we set the knapsack budget to $1/4$. In Figs. 5b and 5d, we compare algorithms based on different values of the knapsack budget, where we consider all the three knapsack constraints. In both of these settings, we again confirm that BARRIER-HEURISTIC, with a very modest computational complexity, outperform state-of-the-art algorithms in terms of the quality of recommended movies.

Figure 5: Movie recommendation: We compare the performance of algorithms over the Movielens dataset. In (a) and (c), we set the knapsack budget to $1/4$. In (b) and (d), we set the maximum cardinality of a feasible solution to 30. We set $\lambda = 0.1$.

## Footnotes

[6]Available for download from: `https://sourceforge.net/projects/louvain/`