[Reviews · NeurIPS 2020]

Review 1

Summary and Contributions: The authors study monotone submodular maximization subject to k-matchoid and \ell knapsack constraints. They use a novel barrier function technique to design a 2(max{k,\ell}+1)-approximation algorithm for the problem that can be modified to achieve an approximation factor of max{k,\ell}+1 at the expense of running time. In particular, they incorporate the knapsack constraints into a potential function that resembles barrier functions used in continuous optimization. The 2(max{k,\ell}+1)-approximation algorithm and one heuristic based on it are experimentally evaluated on various applications related to ML.

Strengths: I really like the main ideas of this paper. This is a classic problem in discrete optimization that has lately found applications in ML and other domains with massive data. Designing faster algorithms with provable guarantees is both well-motivated and relevant to the NeurIPS community. The technique is novel, at least in the discrete domain, and the results seem sound, at least to the extent that I checked.

Weaknesses: The overall impression I have is that this is a very nice theory paper that tries to oversell its practical significance. While the experimental section is as well-written as the rest of the paper and the applications are all interesting, I think that the design of the experiments and its presentation is biased. There are a number of issues; see below.

Correctness: a. I do not see why Barrier-Greedy is not compared to FANTOM [30]. While FANTOM has a worse theoretical guarantee (as it works for non-monotone objectives as well), it is easy to implement, captures a wider range of constraints, it is very fast (using lazy evaluations, it is asymptotically faster than Barrier-Greedy), and it is known to perform extremely well in practice. Since the authors know about [30], it makes no sense that FANTOM was omitted from the comparisons. In fact, along with FAST are the most natural choices to use against Barrier-Greedy. b. The experiments are designed so that r is small and the running time looks linear. In that sense they paint half the picture, as the worst case running time is O(n^3 log n). The reader may be left with the impression that r is always insignificant (this is even highlighted in Footnote 2) and while this is indeed common in practice, it is not universally true. c. While I understand why it makes sense to design Barrier-Heuristic, I do not think using exclusively this in the experiments to be a fair comparison. At least, Barrier-Greedy should be there as well (since this is the main algorithm of this work), no matter how poorly it may perform. In fact, it is quite interesting to see how much of a difference does the heuristic make.

Clarity: The paper is very well-written and the theoretical part was a pleasure to read.

Relation to Prior Work: The paper clearly discusses how it differs from previous contributions.

Reproducibility: Yes

Additional Feedback: -- Parameterizing the running time with the size of a feasible solution is not very common. I know that it has been used before, e.g., in [30], but I think it would be better to state the running times in terms of n and have a corollary about the case r << n. -- As the approximation factors are 2(max{k,\ell}+1) and max{k,\ell}+1, these should be used instead of 2(k+1) and k+1 everywhere, including the abstract. -- l. 70: Isn’t the approximation factor k+1 rather than k? -- l. 88: The statement about the algorithms of Gupta et al. being fast is only partially true. Their algorithm for a knapsack constraint (which is relevant here) is by no means fast. -- l. 135-136: The notation c_{i,a} was not introduced before. -- l. 181-183: the role of \delta_{S,a} is not directly obvious. Maybe elaborate a bit. -- l. 186-187: This is very unclear without reading the pseudocode first. Maybe rephrase. Feel free to respond to any of the above but I would definitely want to see something on my comment about the experiments. ** Update after reading the other reviews and the authors' responses ** I am happy with the authors' responses to the issues I raised.


Review 2

Summary and Contributions: This paper presents a new algorithm based on Barrier functions for constrained submodular maximization. Barrier function based techniques have been extensively used for continuous optimization, but this is the first work to propose such an algorithm for submodular optimization. The authors study the problem of maximizing a monotone submodular function subject to k matchoid constraints and l matroid constraints.

Strengths: Strengths of the Paper: - The authors study the important problem of maximizing a monotone submodular function subject to multiple knapsack and matchoid constraints. - Previous work like the greedy algorithm only works for multiple matroid constraints - Discrete algorithms have always struggled with multiple knapsack constraints, and continuous relaxation based approaches are much more costly. - This is the first work which studied Barrier algorithms for submodular optimization. In my opinion, this is a contribution to the field.

Weaknesses: The main weaknesses which make me a little less excited about this work are mostly related to the scalability of the algorithm(s) and the applications of their techniques. - Scalability: Barrier-Greedy++ is not scalable but achieves a slightly better approximation (by a factor of 2) compared to the relatively more scalable Barrier-Greedy. However, Barrier-Greeedy itself though is linear in n, is cubic in r (which is the size of the largest feasible set). Even if r is a fraction of n (say, 10% of n), the algorithm is still cublc in n which could be very slow for n in range of a million instances. - Motivating applications: Given that greedy algorithm already achieves a factor of k for multiple matroid constraints (which occur mostly in practice), I do not see this algorithm being useful in only the matroid case. Moreover, I do not see any guarantees in the *only* knapsack case (i.e. if there is no matroid constraint). As far as I have seen even in the related work of Mirzasoleiman et al and Feldman et al, they either have only matroid constraints or only knapsack constraints. I would really like the authors to clarify this. In my opining, this paper would be significantly strenghted if this algorithm applies to only multiple knapsack constraints. It seems the algorithm also has an additional assumption of (l \leq k) whcih the authors later relax (but this part was not super clear in the paper).

Correctness: The empirical and theoretical results seem correct

Clarity: For the most part, the paper is well written and clear.

Relation to Prior Work: The authors do a good job comparing to previous work and including all relevant past work. It would help for the authors to add a table contrasting their results with earlier results and comparing the running times.

Reproducibility: Yes

Additional Feedback: See weakness section. Edits after author feedback: After going through the other reviewers feedback and author rebuttal, I think this paper can be accepted for publication at NeurIPS.


Review 3

Summary and Contributions: This paper introduces novel algorithms for submodular maximization based on barrier functions. Particularly, the authors devised two algorithms for monotone submodular maximization subject to k-matchoid and l-knapsack constraints (for l <= k). The first one (BARRIER-GREEDY) gives 2(k+1+ε)-approximation with Õ((nr+r^3)/ε) value oracle calls and Õ(nkr^2/ε) membership oracle calls. The second one (BARRIER-GREEDY++) gives (k+1+ε)-approximation with Õ((n^3r+n^2r^3)ε) value oracle calls and Õ(n^3kr^2/ε) membership oracle calls. The latter algorithm improves the best-known time complexity of [26] even for k-matroids (and no knapsack). The authors complement these theoretical results by real-world experiment, which shows the proposed algorithm (and its heuristic variant) outperforms the baseline methods.

Strengths: - Very strong theoretical result (novel barrier function approach, improved complexity, and approximation ratio matching to the best-known results) - Comprehensive experiment for practical performance

Weaknesses: - BARRIER-GREEDY++ still has higher oracle complexity than baselines in the experiment.

Correctness: I found no mathematical error.

Clarity: Clearly written.

Relation to Prior Work: They compare the proposed methods with known algorithms in detail.

Reproducibility: Yes

Additional Feedback: I enjoyed reading the paper. The barrier function is an old technique in continuous optimization, but it is little used in combinatorial optimization. This paper proposed a novel application of the barrier function method for submodular maximization. The results are strong: BARRIER-GREEDY++ can work with a general class of constraints (k-matchoid and l-knapsack) and yields the best-known approximation ratio. The striking property of their algorithm is the practical usefulness. The previous algorithm either gives a loose approximation ratio or runs heavy local search whose time complexity is too large for almost any real-world ML applications. Adding to these strong theoretical results (I think these theoretical results alone are good enough to accept), the authors conducted a comprehensive experimental evaluation on the performance of their algorithm (and its heuristic variant) in video summarization. Although BARRIER-GREEDY++ has slightly higher oracle complexity than the baselines, the objective value is significantly better. They also designed a heuristic variant of BARRIER-GREEDY, which has better oracle complexity. In conclusion, I believe this paper contributes to both theoretical and practical aspects of submodular optimization and should be accepted. ## update I think the authors successfully addressed the questions in the reviews. I am happy with the authors' feedback.


Review 4

Summary and Contributions: In this paper, the authors solve the problem of constrained maximization of monotone submodular functions with knapsack and matroid/matchoid constraints by constructing a potential function that looks like a barrier function in convex optimization theory. They prove state-of-the-art approximation guarantees, as well as better time complexity. They test their algorithms on two different real-world applications.

Strengths: The paper is well written. All the theorems and lemmas are described properly, and proved correctly and nicely. The approach, up to my knowledge, is novel and combines several ideas from the recent works, as well as bringing new idea of barrier functions into submodular maximization. It is for sure relevant to NeurIPS, especially for people interested in combinatorial/discrete optimization.

Weaknesses: My own appetite for theoretical papers is to have access to "proof sketches" to see how things are proved, instead of putting the whole proof in the appendix. This gives more intuition to the reader, as well as the reviewer. Nevertheless, the page limit prevents most of theoretical works to put in proof sketches; so I consider this not as a weakness, but nice-to-have.

Correctness: The theorems, up to my glance at the appendix seems correct. However, I did not go through the proofs line by line and did not check all the computations.

Clarity: It is indeed a very nice read and I enjoyed a lot.

Relation to Prior Work: Yes, it perfectly fits in the subject.

Reproducibility: Yes

Additional Feedback: ===== EDIT ===== after reading the rebuttal, I keep my score.

[Author Response · NeurIPS 2020]

We thank all the reviewers for their constructive comments. We are also encouraged that the reviewers "like[d] the
main ideas of [our] paper", mentioned that we study an "important problem", stated that is a "very strong theoretical
result" with "comprehensive experiment for practical performance", and added that "the paper is well written". In this
response, we address the specific questions asked by each reviewer one by one.

**Reviewer #1 Q. Why Barrier-Greedy is not compared to FANTOM?** We had performed the first set of experiments
where FANTOM was included. We decided to exclude FANTOM for two main reasons: (i) The first iteration of
FANTOM is similar to FAST, where FANTOM uses an iterated greedy algorithm with density threshold to guarantee
the performance of the algorithm for non-monotone submodular functions. As a result, the performance of the two
algorithms would be quite close for monotone submodular functions (See Figure 1a and 1c). (ii) Due to the several
iterations of the iterated greedy in FANTOM, its computational complexity highly increases (See Figures 1b and 1d)
without any gained benefit for the utility. Given these, we thought that for a more fair comparison it is better to not
report the performance of FANTOM. Definitely, in the revised version, we can report the results of FANTOM for all
    our experiments.

(a) Figure 1a of the paper    (b) Figure 1c of the paper    (c) Figure 2a of the paper    (d) Figure 2c of the paper

Figure 1: A few sample experiments from the paper where FANTOM is included.

**Reviewer #1 Q. The experiments are designed so that r is small and the running time looks linear.** We agree that
generally $r$ could be in the order of $n$. In that case, the term $r^3$ dominates the term $nr$ and the linearity argument is not
valid anymore. We will clarify this in the revised version.

**Reviewer #1 Q. Compare Barrier-Heuristic with Barrier-Greedy in the experimental section.** We will add the ex-
perimental results for Barrier-Greedy in applications with more than one knapsack constraint. In our initial experiments,
we do not observe a dramatic reduction and Barrier-Greedy is at least as good as the other baseline algorithms.

**Reviewer #1 Q. Parameterizing the running time with the size of a feasible solution is not very common.** We
will state the time complexities in terms of $n$ and will add a corollary for cases with $r \ll n$.

**Reviewer #2 Q. This algorithm is not useful in only the matroid case. Moreover, there is no guarantee in the
only knapsack case.** We should point out that the power of our algorithm is in solving the monotone submodular
maximization problem subject to the intersection of these two constraints. Indeed, the guarantee of our algorithm for
only $k$-matroids is $(k + 1)$ (which is slightly worse than greedy). For the case of only $\ell$ knapsacks, the approximation
factor is $\ell + 1$. We will add this discussion to the revised version of the paper.

**Reviewer #2 Q. Importance of the constraints studied in this paper.** We believe the problem of maximizing a
submodular function subject to the interaction of $k$-matroids and $\ell$-knapsack constraints has both theoretical and
practical importance. Indeed, in many optimization tasks that are subject to some structural constraints, we have a
limited budget. The combination of matroids and knapsack constraints is a natural way to model these problems. Along
with our work, Mirzasoleiman et al. [30] used the same set of constraints to model recommendation systems, image
summarization, and revenue maximization tasks.

**Reviewer #2 Q. Add a table contrasting their results with earlier results and comparing the running times.** We
will add a table to compare our results with previous works.

**Reviewer #2 Reviewer #3 Q. Scalability of Barrier-Greedy and Barrier-Greedy++.** Although our algorithms are
not completely suitable for very large scale problems (at least theoretically), we provide the state of the art result that
lies in between fast algorithms with suboptimal approximation ratios and slower algorithms. We should mention that
Barrier-Greedy and Barrier-Heuristic algorithms are quite fast in practice (they are comparable with FAST) and this
makes them applicable in practical scenarios. We also agree that Barrier-Greedy++ is not scalable enough to be used for
large scale applications. The main purpose of Barrier-Greedy++ is to provide a tighter approximation guarantee at the
expense of a higher computational cost. Indeed, there is a trade-off between the quality of the solution we desire to
obtain (higher for Barrier-Greedy++) and the time we are willing to spend (faster for Barrier-Greedy).

**Reviewer #4 Q. Access to "proof sketches" to see how things are proved, instead of putting the whole proof in
the appendix.** We will add a more detailed proof sketch to the main text.

[Meta-Review · NeurIPS 2020]

This was a unanimous accept.